# Graphene Oxide Thin Films for Detection and Quantification of Industrially Relevant Alcohols and Acetic Acid

**DOI:** 10.3390/s23010462

**Published:** 2023-01-01

**Authors:** Pedro Catalão Moura, Thais Priscilla Pivetta, Valentina Vassilenko, Paulo António Ribeiro, Maria Raposo

**Affiliations:** Laboratory for Instrumentation, Biomedical Engineering and Radiation Physics (LIBPhys-NOVA), Department of Physics, NOVA School of Science and Technology, NOVA University of Lisbon, Campus FCT-NOVA, 2829-516 Caparica, Portugal

**Keywords:** volatile organic compounds, VOC, industrial environment, indoor air, air quality, acetic acid, ethanol, methanol, isopropanol, electronic nose, impedance spectroscopy, layer-by-layer films

## Abstract

Industrial environments are frequently composed of potentially toxic and hazardous compounds. Volatile organic compounds (VOCs) are one of the most concerning categories of analytes commonly existent in the indoor air of factories’ facilities. The sources of VOCs in the industrial context are abundant and a vast range of human health conditions and pathologies are known to be caused by both short- and long-term exposures. Hence, accurate and rapid detection, identification, and quantification of VOCs in industrial environments are mandatory issues. This work demonstrates that graphene oxide (GO) thin films can be used to distinguish acetic acid, ethanol, isopropanol, and methanol, major analytes for the field of industrial air quality, using the electronic nose concept based on impedance spectra measurements. The data were treated by principal component analysis. The sensor consists of polyethyleneimine (PEI) and GO layer-by-layer films deposited on ceramic supports coated with gold interdigitated electrodes. The electrical characterization of this sensor in the presence of the VOCs allows the identification of acetic acid in the concentration range from 24 to 120 ppm, and of ethanol, isopropanol, and methanol in a concentration range from 18 to 90 ppm, respectively. Moreover, the results allows the quantification of acetic acid, ethanol, and isopropanol concentrations with sensitivity values of $\left(3.03\pm 0.12\right)*10^{4}$, $\left(-1.15\pm 0.19\right)*10^{4}$, and $\left(-1.1\pm 0.50\right)*10^{4}$ mL^−1^, respectively. The resolution of this sensor to detect the different analytes is lower than 0.04 ppm, which means it is an interesting sensor for use as an electronic nose for the detection of VOCs.

## 1. Introduction

The indoor environment of industrial facilities, particularly in production lines and warehouses, is commonly populated by a large variety of potentially polluting and hazardous compounds [1,2,3]. Their presence in the air arises from the multitude of emitting sources existent in these kinds of scenarios and leads to well-known and worthy-of-attention consequences for the employee’s health [4,5]. Volatile organic compounds (VOCs) are among the most concerning of these potentially hazardous analytes.

Volatile organic compounds correspond to organic compounds whose vapor pressure, at 293.15 K, equals or exceeds 10 Pa, i.e., they are volatile at room temperature [6]. Considering their nature, VOC-emitting sources can be arranged into two distinct categories, natural sources and anthropogenic sources. Natural sources include fauna and flora emissions. Smoking, cooking, or cleaning, and all the products related to these activities, such as tobacco, food, perfumes, personal care creams, and detergents, for example, are among the main anthropogenic sources of VOCs. A vast range of daily-use objects, namely clothes, furniture, building materials, paints, fuels, sprays, pesticides, glues, writing materials, and copying devices, are equally relevant anthropogenic sources of VOCs [2,7,8].

At an industrial level, VOCs are a common element of the indoor air of the facilities since the activities that are usually undertaken in such locations are conducive to the emission of these kinds of analytes. For instance, in coating industries and facilities with painting, printing, or similar activities, it is rather common for the detection of relevant amounts of alcohol-based VOCs in the atmosphere due to the frequent use of solvents, paints, and other coating solutions [9,10,11]. Automotive, electronics, and comparable assembly lines are equally replete with numerous sources of VOCs, namely, the chemicals, solvents, or rubbers, and the welding, drying, heating, and coating processes often employed during the production [3,12,13]. The manufacturing facilities of personal care and cleaning products, due to the intense utilization of VOCs-based chemicals in their formulas, are major contributors to the presence of VOCs in both indoor and outdoor air [14,15,16]. Summarizing, independently of the undertaken activities, industrial facilities are often crowded by sources of VOCs such as acetic acid, ethanol, isopropanol, methanol, and many others.

Exposure to VOCs in both short- and long-term scenarios is known for causing an extensive list of pathologies and health conditions that ranges from harmless biological reactions to health-threatening diseases [17]. In simpler cases, exposure to VOCs leads to allergic or inflammatory reactions in the respiratory tract, and cutaneous and ocular tissues. Headaches, nausea, dizziness, visual disorders, memory impairment, emesis, epistaxis, and fatigue are equally ordinary and well-known reactions of the human organism to the presence of VOCs [18,19]. A cause–consequence relation between continued exposure to VOCs and the development of dangerous forms of cancer has equally been studied. Lung, oral, and even breast cancer are examples of VOC-related carcinogenic pathologies [20,21].

Due to all the aforementioned facts, it is mandatory to study, develop, and implement analytical tools that enable the accurate and rapid detection, identification, and quantification of the presence of VOCs in the indoor air of industrial facilities and, consequently, the prevention of potential hazards to both the environment and employees’ health.

Several techniques have been scientifically addressed regarding their suitability for the assessment of VOCs. These techniques include both multisensor array-based procedures and analytical techniques such as chromatographic and spectrometric approaches. Independently of the designation or the nature of the system, their core purpose is the detection of specific and potentially dangerous analytes.

Regarding chromatographic and spectrometric systems, their suitability for the detection of VOCs has been largely investigated in a substantial amount of practical applications that include air quality assessment, safety conditions, food characterization, drug detection, clinical scenarios, and many others [1,22,23,24]. The main advantages of analytical techniques such as liquid chromatography (LC), gas chromatography (GC), mass spectrometry (MS), or ion mobility spectrometry (IMS) include their high levels of sensitivity and precision, wide dynamic concentration ranges, analytical flexibility, and almost real-time monitoring capability. Notwithstanding these advantages, they also have some limitations, namely, their lack of portability and high costs, the necessity for sample preparation, and the requirement for qualified personnel [25,26,27].

Aiming to circumvent the mentioned limitations, the development of sensor array-based systems dedicated to specific scenarios has gained relevance as a cheaper and simpler solution [28,29]. These systems enable an accurate and rapid qualification and quantification of VOCs through the interactions that occur on the surface of the sensor when they experience contact with the analyte [30,31]. Due to their flexibility and scientifically relevant results, sensor array-based systems have proved their suitability for the assessment of common VOCs such as acetone [32], ethanol [33], butanol [34], formaldehyde [35], triethylamine [36], methanol [37], isopropanol [38], ethyl acetate [39], benzene [40], or acetic acid [41], among many others.

The development of electronic tongues [42,43,44] and noses [30] based on featured organic thin films of graphene as sensing units have been widely addressed over the past years; an approach that can also be used for the development of VOC-dedicated electronic noses and tongues. For these cases, a sensor array is normally required. Graphene molecules and derivatives were shown to be suitable to be used to detect a panoply of molecules, macromolecules, and even viruses; thus, making them an invaluable tool in many fields (e.g., medicine, industry, genetics, criminology) [45]. Moreover, graphene oxide (GO) can be used in both electrical and optical devices [46]. Graphene oxide and poly (allylamine hydrochloride) (PAH) layer-by-layer thin films (LBL) have been characterized with respect to their growth with the number of bilayers, morphology, and electrical properties [47]. The electrical characterization of these PAH/GO films revealed a semiconductor behaviour that makes these films interesting for the development of sensors by probing their electrical property changes when submitted to different environments [47]. This work proposes the development of a graphene oxide thin-film-based sensor using the layer-by-layer technique, towards the detection, identification, and quantification of industrially relevant VOCs, namely acetic acid, ethanol, isopropanol, and methanol. Impedance spectroscopy was used as a probe of the sensor response in terms of the analyte and concentration and data were processed through principal component analysis (PCA). The achieved results revealed the potential of the sensor being used not only to discriminate these compounds in a complex mixture but also to quantify them, which could be a factor that adds value towards its use for air quality control and public health.

## 2. Materials and Methods

### 2.1. Materials

Polyethyleneimine (PEI) and graphene oxide (GO), utilized for the preparation of thin films, were purchased from Sigma-Aldrich. Standards of acetone (C_3_H_6_O; 58.08 gmol^−1^; 99.0%) and isopropanol (C_3_H_8_O; 60.10 gmol^−1^; 98.0%) were purchased from Laborspirit-Labchem. Standards of ethanol (C_2_H_6_O; 46.07 gmol^−1^; 99.8%) and methanol (CH_4_O; 32.04 gmol^−1^; 99.8%) were obtained from Honeywell. Standard of acetic acid (C_2_H_4_O_2_; 60.05 gmol^−1^; 99.8%) was purchased from Fisher Scientific. The ceramic-based sensor supports with deposited gold interdigitated electrodes (IDE) were acquired from Metrohm DropSens (length: 22.8 mm; width: 7.6 mm; thickness: 1 mm; electrodes width: 200 μm; distance between electrodes: 200 μm).

### 2.2. Preparation of Sensor

Thin films of PEI and GO polyelectrolytes were adsorbed by layer-by-layer (LbL) technique on ceramic-based sensor supports with deposited gold IDE. This technique basically settles down on electrostatic forces of the polyelectrolytes that enable the application of alternated electrically charged polyelectrolytes [48]. To do so, the supports were alternatively immersed in the PEI and GO aqueous solutions, positively and negatively charged polyelectrolyte solutions with concentrations of 2.5 × 10^−1^ and 3.2 × 10^−1^ mg/mL, respectively. Between each immersion, a wash procedure, consisting of the immersion of the support in ultrapure water to remove eventual excesses of polyelectrolyte, was undertaken. Once completing each immersion sequence, the support was dried with a gentle nitrogen air blasting. The described procedure corresponds to the deposition of a bilayer and was repeated 15 times, leading to the deposition of the multilayer thin films of polyelectrolyte in the surface of the ceramic-based support, and forming the sensors denominated as (PEI/GO)_15_. Detailed information regarding the procedure was previously described in the literature [49,50].

### 2.3. Impedance Spectroscopy

Impedance spectroscopy was the analytical technique selected for the characterization of sensor units’ response to the industrial solvents mentioned above. Detailed information on experimental setup has been described by Magro et al. [30]. Basically, a custom-made chamber with a volume of 58 L was employed to create a controlled atmosphere and assess the electrical impedance response of the thin films. Initially, the chamber was evacuated to pressure of 10^−3^ mbar and the previously calibrated sample of the target VOCs was introduced into a round-bottom glass flask. The VOC sample was then volatilized and purged through compressed synthetic air (ALPHAGAZ^TM^ 1 AR—Air Liquide). The sensor units were previously placed in the respective sample holder in the chamber and, as next step, their electrical response to the VOCs was measured with an impedance analyzer (Solartron 1260 Impedance/Gain-Phase Analyzer coupled with a 1296A Dielectric Interface-AMETEK Scientific Instruments) and assessed with a dedicated software (SMaRT Impedance Measurement Software, version 3.3.1-AMETEK Scientific Instruments). A frequency range of 1 to 10^6^ Hz and an AC voltage of 25 mV were applied during the impedance assessment. To ensure that the signal is representative of the sensor’s response, these measurements were performed in triplicate.

### 2.4. Data Treatment

Principal component analysis (PCA) was used to reduce the data size and to obtain a new space of orthogonal components aiming to distinguish the different samples and respective concentrations [51]. For this analysis, the electrical impedance and impedance angle spectra were considered. It is relevant to emphasize that these values were both collected in a frequency range of 1 Hz to 1 MHz for each sample at different concentrations. Since three replicas were registered, the spectra used in this analysis correspond to the average of those three measurements.

## 3. Results

### Impedance Results

Figure 1 illustrates the impedance (a) and impedance angle (b) spectra of (PEI/GO)_15_ films deposited on the surface of gold IDEs when submitted to an atmosphere of different concentrations of acetic acid (I), ethanol (II), methanol (III), and isopropanol (IV), represented by the VOCs’ evaporated volume. It should be mentioned that, for better clarity and interpretation of the plots present in the figure, the error bars measured for these spectra were not included. Nonetheless, it is relevant to state that both these frequency-dependent measurements presented error values lower than 1% among the three replicas.

To verify if the PEI/GO thin films can distinguish the different concentrations of the measured VOCs, the impedance magnitude and impedance angle at fixed frequencies were analyzed for the different VOCs’ concentrations. The evaporated volume can be directly related with the concentration levels through the ratio between the mass of the analyte and the mass of the air in the chamber. The mass of each analyte was calculated for each volume through their density. The same approach was employed to calculate the mass of the air existent in the interior of the 58 L volume chamber. Once both the analyte mass for each volume and the total mass in the interior of the chamber were calculated, the corresponding concentrations were estimated by the mentioned ratio. Table 1 summarizes the evaporated volumes and respective concentration levels for the four analysed analytes. The concentration levels were converted to ppm scale for easier comprehension.

Figure 2a–d show both the impedance magnitude and impedance angle at 10^4^ Hz, plotted as a function of the evaporated volume and, consequently, the concentration of acetic acid, ethanol, methanol, and isopropanol, respectively. These graphs clearly demonstrate that the electrical measurements can distinguish between the different concentrations since both the magnitude and angle vary with the concentration if only a VOC type is considered. However, when analyzing the measured values of magnitude and angle at this chosen frequency, one cannot distinguish between the different alcohol VOCs, meaning that is necessary to analyze the data achieved for all the frequencies with mathematical methods such as the PCA method. This analysis allows conclusions to be made as to whether the impedance magnitude and impedance angle spectra depend on the different samples and allows discrimination between the VOCs and their respective concentrations using a single sensor.

## 4. Discussion

As one intends to distinguish between different VOCs using a single sensor, the PCA method was applied to both the impedance magnitude and impedance angle spectra data measured for the different concentrations of acetic acid, ethanol, methanol, and isopropanol. The PCA score plots of all the measured data for all the four target analytes in air at different concentrations are shown in Figure 3a. By analyzing this figure, one can observe that well defined PCA score regions can be defined for each type of measured VOC, allowing discrimination between the samples in the concentrations measured using a single sensor.

To analyze if the calculated principal components PC1 and PC2 are concentration-dependent, the achieved values of PC1 and PC2 were plotted as a function of the VOC’s volume in Figure 3b,c, respectively. Interestingly, the evolution of PC1 with the solvent volume shows that, except for the PC1 value associated to samples without VOCs and for the acetic acid sample with a volume of 200 μL, one can calculate a PC1 average value for each type of sample. The calculated values and respective error bars are present in Table 2. Therefore, it is possible to state that the PC1 value can distinguish the ethanol, methanol, and isopropanol in air samples and the acid acetic for higher concentrations. On the other hand, Figure 3c clearly demonstrates that the PC2 components can discriminate the VOCs’ concentrations, with methanol being the exception. The sensitivity of the sensor when submitted to the different VOCs was estimated by fitting the PC2 data versus concentration with a straight line. The fitting parameters are listed in Table 2, where the sensitivity values correspond to the slope of the PC2 parameters versus the concentration. Sensitivity values of 30,300 ± 1200, −11,500 ± 1900, and −11,000 ± 5000 mL^−1^ were calculated for acetic acid, ethanol, and isopropanol, respectively. Since the slope calculated for the case of methanol PC2 data is very low, one decided to calculate the PC2 average. All these values are displayed in Table 2. From these results, one can conclude that this unique sensor can distinguish the different target VOCs and, except for methanol, the VOCs’ concentration in the analyzed range.

To compare the achieved data with the results of other sensors existent in the literature, one presents Table 3 where the values of the resolution and range values of the developed sensor are compared with the values of different sensors. To calculate the sensor resolution for this work, the minimum measurable values were considered, as described elsewhere [52]. As in the present results, the principal component 2 (*PC2*) values are linearly dependent on the concentration in ppm (*C*). The sensitivity (*S*) was calculated by the slope of the straight line, ΔPC2/ΔC, used to fit the data. Therefore, ΔPC2/ΔC=S±u(S), with *u(S)* being the uncertainty of sensitivity given also by the fitting. The resolution corresponds to the calculated value of ∆*C* in which ΔC=u(S)/S. This procedure enabled the estimation of the resolution values of 0.005, 0.015, and 0.04 ppm for acetic acid, ethanol, and isopropanol, respectively. As aforementioned, due to the very low slope calculated for methanol, one opted to not estimate the sensitivity and, consequently, the resolution values of this case. From the comparison included in Table 3, one can conclude that the methodology described in this work leads to limited resolution even though the studied range is of an intermediate level. The achieved values of resolution indicate that this sensor can be used in the development of an electronic nose for the detection of VOCs.

## 5. Conclusions

A unique sensor based on GO oxide thin films was used to simultaneously detect four industrially relevant VOCs, acetic acid, ethanol, methanol, and isopropanol, by measuring the impedance magnitude and impedance angle spectra responses in terms of concentrations and processing the data through PCA. The results lead to the conclusion that the impedance data allow both the different VOCs samples and their concentrations to be distinguished in the range of hundreds of ppm. From the PCA results, one can conclude that the principal component PC1 values can distinguish the ethanol, methanol, and isopropanol in air samples and also the acid acetic for higher concentrations, while from principal component PC2, one can discriminate the VOCs’ concentrations with the exception of methanol. The sensitivities of the sensor are 30,300 ± 1200, −11,500 ± 1900, and −11,000 ± 5000 mL^−1^ for the acetic acid, ethanol, and isopropanol, respectively. The resolution values for this sensor are lower than 0.04 ppm, which proves the relevancy of using this sensor in the sensor array of an electronic nose for the qualification and quantification of VOCs.

## Figures and Tables

**Figure 1 sensors-23-00462-f001:**
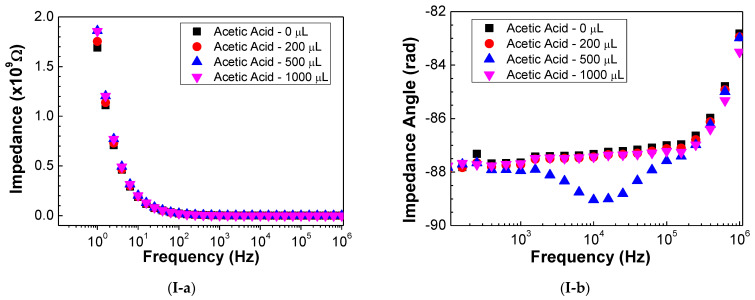
Impedance (**a**) and impedance angle (**b**) spectra of the sensor devices when exposed to atmospheres with different concentrations of acetic acid (**I**), ethanol (**II**), methanol (**III**), and isopropanol (**IV**).

**Figure 2 sensors-23-00462-f002:**
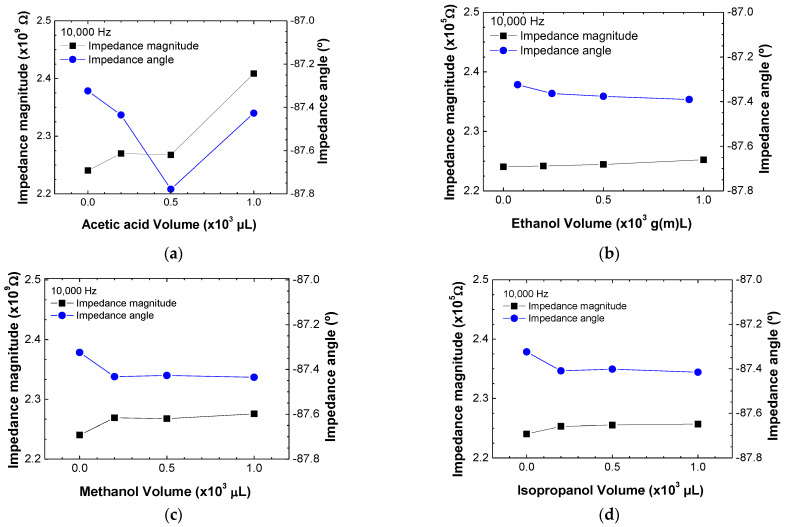
Impedance magnitude and impedance angle at a fixed frequency of 10^4^ Hz for different concentrations of acetic acid (**a**), ethanol (**b**), methanol (**c**), and isopropanol (**d**) in air. The lines between the experimental points are guidelines.

**Figure 3 sensors-23-00462-f003:**
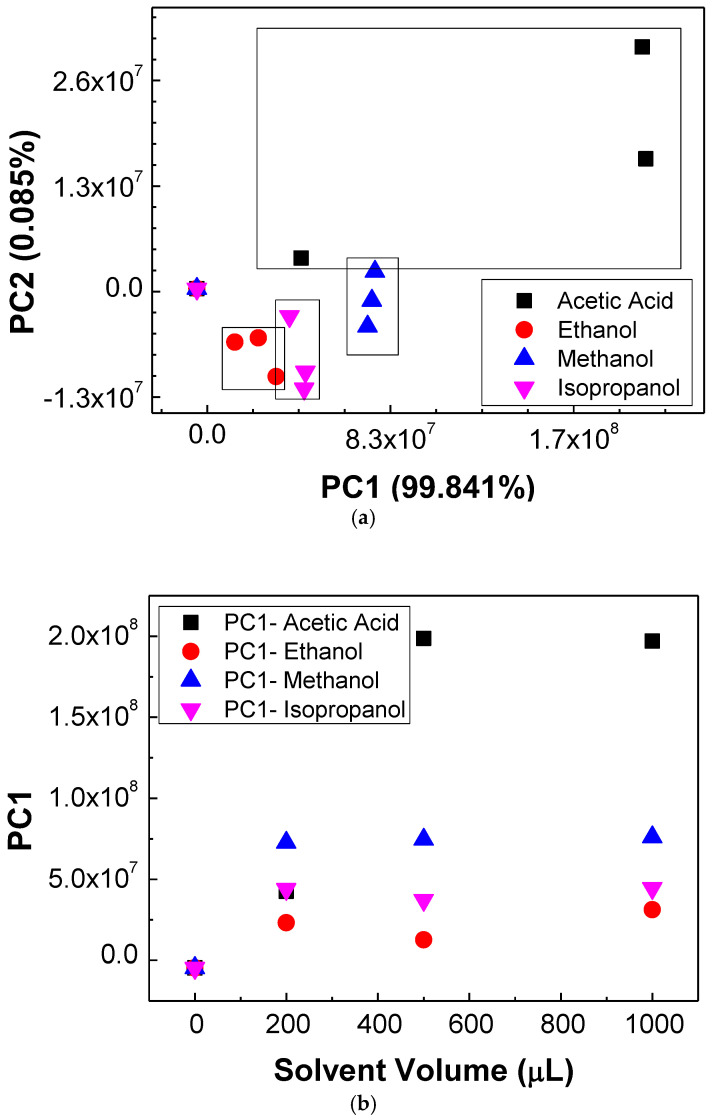
(**a**) PCA score plot after analyzing all the measured data for the detection of acetic acid, ethanol, methanol, and isopropanol in air at different concentrations; (**b**) evolution of PC1 components as a function of solvent volume; (**c**) evolution of PC2 components as a function of solvent volume.

**Table 1 sensors-23-00462-t001:** Evaporated volumes (μL) and respective concentration levels (ppm) for the four considered VOCs; acetic acid, ethanol, methanol, and isopropanol.

Volume (μL)	Concentration (ppm)			
	Acetic Acid	Ethanol	Methanol	Isopropanol
200	24	18	18	18
500	60	45	45	45
1000	120	90	90	90

**Table 2 sensors-23-00462-t002:** Summarization of the values achieved from Figure 3b,c. The columns PC1 Average and PC2 Average represent the mean value calculated from data of Figure 3b,c for the case of methanol, not considering the null concentration data. The sensitivity corresponds to the slope of the straight lines fitting PC2 data displayed in Figure 3c.

VOC	PC1 Average	Sensitivity (mL^−1^)	PC2 Average
Acetic acid	(1.98 ± 0.01) × 10^8^ *	(3.03 ± 0.12) × 10^4^	-
Ethanol	(2.23 ± 0.54) × 10^7^	(−1.15 ± 0.19) × 10^4^	-
Methanol	(7.4 ± 0.1) × 10^7^	-	(−0.066 ± 2.81) × 10^6^
Isopropanol	(4.19 ± 0.23) × 10^7^	(−1.1 ± 0.5) × 10^4^	-

* For concentrations higher than or equal to 60 ppm.

**Table 3 sensors-23-00462-t003:** Comparison of the achieved sensors with others available in literature.

Sensor	Resolution (ppm)	Range (ppm)
Acetic acid [53]	1.2	1–13
Acetic acid [54]	1	10–100
Acetic acid [55]	0.5	0.5–2000
Acetic acid [56]	0.73	1–15
Acetic acid (this work)	0.005	24–240
Ethanol [57]	0.05	1–200
Ethanol [58]	3	30–145
Ethanol [59]	1	1–200
Ethanol [60]	0.15	0.15–5
Ethanol (this work)	0.015	18–180
Methanol [61]	0.015	1.14–11.36
Methanol [62]	10	100–300
Methanol [63]	0.5	0.5–700
Methanol [64]	10	100–500
Methanol (this work)	-	18–180
Isopropanol [65]	2	2–100
Isopropanol [66]	1	1–100
Isopropanol [67]	1	1–1000
Isopropanol [68]	1	5–1000
Isopropanol (this work)	0.04	18–180

## Data Availability

Datasets are available from the corresponding authors upon reasonable request.

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
