# Peer review of "Graphene Oxide Thin Films for Detection and Quantification of Industrially Relevant Alcohols and Acetic Acid"

_sensors, 2023, doi:10.3390/s23010462_

Round 1
Reviewer 1 Report
Through this manuscript entitled "Graphene Oxide Thin Films for Detection and Quantification of Industrially Relevant Alcohols and Acetic Acid" the authors present a type of gas sensor based on graphene oxide (GO) thin films that can be used to distinguish acetic acid, ethanol, isopropanol and methanol, major analyses in the field of industrial air quality and used the electronic nose concept based on impedance spectra measurements.
The topic of the manuscript is interesting and topical, taking into account the fact that it is desired to create a multicomponent sensor, capable of detecting several types of gases in the air. However some comments are given below:
-
The legend from Figure 1 should be written in the same font, all in upper or lower case and in the same language.
-
Figure 2 presents the impedance magnitude and the impedance angle at a frequency of 10000 Hz for different concentrations of acetic acid, ethanol, methanol, and isopropanol (d) in air, but on the x axis is presented the VOC volume. What are the concentrations used?
-
What GO concentrations were used to optimize the material properties?
-
The thin films were analyzed usng standard measurments techniques such as FTIR, XRD, Raman, XPS, TEM or SEM?
-
What are the sensing properties such as response time and recovery time of sensor?
-
In the Conclusions section, it is said that the obtained results distinguish between different concentrations of VOCs. However, the manuscript presents results related to the volume of VOCs and not to different concentrations of the concentration. Have you determined the response of the sensor to different concentrations of VOCs?

Author Response
Reviewer#1
Through this manuscript entitled "Graphene Oxide Thin Films for Detection and Quantification of Industrially Relevant Alcohols and Acetic Acid" the authors present a type of gas sensor based on graphene oxide (GO) thin films that can be used to distinguish acetic acid, ethanol, isopropanol and methanol, major analyses in the field of industrial air quality and used the electronic nose concept based on impedance spectra measurements. The topic of the manuscript is interesting and topical, taking into account the fact that it is desired to create a multicomponent sensor, capable of detecting several types of gases in the air. However some comments are given below:
Answer: Dear reviewer, we are glad that the main goals of our most recent research paper were clear to you. Thank you for your fair evaluation of our work and for all your useful suggestions. Every single comment was addressed, and corresponding corrections were implemented. We hope that the new version of the manuscript meets your expectations.
The legend from Figure 1 should be written in the same font, all in upper or lower case and in the same language.
Answer: We would like to apologize for this incoherence in the language and font used in the figures. Corrections were implemented. Thank you for alerting us about this issue.
Figure 2 presents the impedance magnitude and the impedance angle at a frequency of 10000 Hz for different concentrations of acetic acid, ethanol, methanol, and isopropanol (d) in air, but on the x axis is presented the VOC volume. What are the concentrations used?
Answer: Additional information and a table were included in this section of the manuscript aiming to clarify the concentration levels corresponding to the evaporated volumes for each one of the four analytes. Regarding the values, 200 μL corresponded to 18 and 24 ppm respectively for the three alcohols and acetic acid; 500 μL corresponded to 45 and 60 ppm respectively for the three alcohols and acetic acid; 1000 μL corresponded to 90 and 120 ppm respectively for the three alcohols and acetic acid. We hope that, by adding this additional information, we were able of improving the clarity of the manuscript.
What GO concentrations were used to optimize the material properties?
Answer: Thank you for your question, which we are happy to reply. In the preparation of GO thin films, the GO aqueous solutions used presented a concentration of 0.32mg/mL. This concentration resulted from previous works, as addressed in the manuscript, namely the works cited as:
[49] - Zagalo, P.M.; Ribeiro, P.A.; Raposo, M. Effect of Applied Electrical Stimuli to Interdigitated Electrode Sensors While Detecting 17a-Ethinylestradiol in Water Samples. Chemosensors 2022, 10, 114.
[50] - Zagalo, P.M.; Ribeiro, P.A.; Raposo, M. Detecting Traces of 17a-Ethinylestradiol in Complex Water Matrices. Sensors 2020, 20, 7324.
The thin films were analyzed usng standard measurments techniques such as FTIR, XRD, Raman, XPS, TEM or SEM?
Answer: Thank you for this question, which we are happy to answer. These GO thin films have been analysed with different techniques. One example of those techniques is the vacuum ultraviolet spectroscopy and atomic force microscopy, as can be consulted in the work cited in the bibliography as:
[47] - Assunção, I.C.C.; Sério, S.; Ferreira, Q.; Jones, N.C.; Hoffmann, S.V.; Ribeiro, P.A.; Raposo, M. Graphene Oxide Layer-by-Layer Films for Sensors and Devices. Nanomaterials 2021, 11, 1556.
What are the sensing properties such as response time and recovery time of sensor?
Answer: As pointed by the reviewer, this work does not include information on the response and recovery times of the sensor since they were not estimated. Nonetheless, it is relevant to state that the described procedure was carefully and repeatedly employed for every single analysed sample. Between each sample, the chamber was evacuated until a pressure of 10-3 mbar and the chamber fulfilled with the next sample which means that the time between measurements is practically constant. Even at the end of the day the chamber was evacuated after the last measurement but in this case the next sample was introduced in the chamber in the next day. The procedure guaranteed the repeatability of the results throughout all the analyses done during this research work. Therefore the response time and recovery time were not studied yet.
In the Conclusions section, it is said that the obtained results distinguish between different concentrations of VOCs. However, the manuscript presents results related to the volume of VOCs and not to different concentrations of the concentration. Have you determined the response of the sensor to different concentrations of VOCs?
Answer: Yes, the response of the sensors was tested for different concentrations of all the four VOCs. As I mentioned in one of my previous comments, a table was added to the manuscript in order to clarify the considered concentration levels relatively to the evaporated volumes. Briefly, 200 μL corresponded to 18 and 24 ppm respectively for the three alcohols and acetic acid; 500 μL corresponded to 45 and 60 ppm respectively for the three alcohols and acetic acid; 1000 μL corresponded to 90 and 120 ppm respectively for the three alcohols and acetic acid. Considering these values, it is safe to state that we successfully tested the response of our sensors to different concentrations. Thank you for alerting us about the lack of clarity in the section of the manuscript.

Reviewer 2 Report
The paper studies gas sensors based on polyethyleneimine (PEI) and GO films deposited on ceramic supports. The film is deposited by a layer-by-layer procedure, already presented in the past. The VOC sensor is a chemiresistor, and by evaluating the impedance at different frequencies, the Authors defined the sensor response to different analytes as a function of their concentration. The results enlighten the possibility of using PEI/GO as an active sensing element. However, the presented data suggest a poor selectivity, in some cases a poor dependence vs. analyte concentration but a final state-of-the-art resolution. These inconsistencies must be solved, also by a more accurate data analysis. Several issues must be addressed before publication. In more detail:
1. Introduction: lines 35-38 must be placed at the end of this section, i.e., after line 121. The introduction is about one-third of the paper text; it is complete but oversized and must be reduced.
2. Materials: line 150, it needs to be clarified when the Authors apply the drying procedure, as according to ref 46 it is done only after the completion of each bilayer. The description must be improved.
3. Results: the claim in lines 201-203 is unclear and untrue. Figures 2 c and d show the sensor's difficulty distinguishing between different concentrations, not their ability.
4. Error evaluation: the Authors claim about a 1% or less error bar (lines 188-190). For an impedance of 2x10^5 ohm, the error is 2x10^3 ohm, and Fig.2 left axes often do not enable the minimum step identification; for an angle of about 80°, the error is about 0.8°, thus, more significant than the detected differences of about 0.1°. Moreover, the impedance magnitude and angle values at zero analyte concentration (that should be the same, within errors) in Fig.2 and 3 suggest a significant uncertainty larger than 1%.
5. PCA analysis: results in table 1 are shown in normal and scientific notation, making every comparison difficult. The Authors should describe how they evaluated the ppm concentration (starting from the solvent volume) and the sensor resolution, all quantities that compare in Table 2. PCA score plot does not show “well defined” regions; in fact, the Authors analyzed each PC role to understand that PC1 distinguishes the different analytes and PC2 can somehow evaluate a sort of sensitivity. The Authors must claim the difficulties in evaluating the VOC concentrations.
6. Resolution: the reported values are state-of-the-art, compared with other results, but the Authors do not emphasize this result and must explain how they found these values (see also point 4 and 5).
7. Figure 1: several legends are not written in English and use different styles (spaces, uppercase, normal/bold text), same for axis labels. uL should be µL. Figure IVa and b show different isopropanol concentrations.
8. Figure 2: axis labels have different styles. For b, c, and d left scales, it is impossible to identify the minimum step.
9. Figure 3: legends should use the same nomenclature as in previous figures, i.e., Ethanol and not EtOh.
10. English quality must be improved, and several typos removed.
Author Response
We thank to the Editor and to the Reviewers for the comments and suggestions that we think are properly addressed in the present form of the manuscript. We think that our address to these comments, prompted by this review, improved its overall quality.
Reviewer #2
The paper studies gas sensors based on polyethyleneimine (PEI) and GO films deposited on ceramic supports. The film is deposited by a layer-by-layer procedure, already presented in the past. The VOC sensor is a chemiresistor, and by evaluating the impedance at different frequencies, the Authors defined the sensor response to different analytes as a function of their concentration. The results enlighten the possibility of using PEI/GO as an active sensing element. However, the presented data suggest a poor selectivity, in some cases a poor dependence vs. analyte concentration but a final state-of-the-art resolution. These inconsistencies must be solved, also by a more accurate data analysis. Several issues must be addressed before publication. In more detail:
- Introduction: lines 35-38 must be placed at the end of this section, i.e., after line 121. The introduction is about one-third of the paper text; it is complete but oversized and must be reduced.
Answer: Thank you for the suggestion. The introduction was reduced.
- Materials: line 150, it needs to be clarified when the Authors apply the drying procedure, as according to ref 46 it is done only after the completion of each bilayer. The description must be improved.
Answer: We recognize that this portion of the text could be ambiguous. Corrections aiming to increase the text clarity were implemented. Thank you for alerting us about this issue.
- Results: the claim in lines 201-203 is unclear and untrue. Figures 2 c and d show the sensor's difficulty distinguishing between different concentrations, not their ability.
Answer: Thank you very much for this important comment. In fact, we are interested to demonstrate that for a distinct VOC the electrical measurements can indicate the sample concentration. But in fact the data also show that at this frequency one cannot distinguish clearly between the different alcohols samples. The manuscript was improved to better describe the experimental results and the scale of the figures were changed in order to clarify those conclusions.
- Error evaluation: the Authors claim about a 1% or less error bar (lines 188-190). For an impedance of 2x10^5 ohm, the error is 2x10^3 ohm, and Fig.2 left axes often do not enable the minimum step identification; for an angle of about 80°, the error is about 0.8°, thus, more significant than the detected differences of about 0.1°. Moreover, the impedance magnitude and angle values at zero analyte concentration (that should be the same, within errors) in Fig.2 and 3 suggest a significant uncertainty larger than 1%.
Answer: Regarding the size of the axes, all the figures/graphs were fixed in order to attain all the recommendations of the reviewer. Thank you for alerting us about this issue. The 1% value addressed in the first paragraph of the Results section is regarding the repetition of the measurements. As mentioned in the manuscript, the analyses were replicated three times to assess the repeatability of the procedure. It was possible to see that the variation of the data collected among the three replicas stayed under 1%, proving that not only the procedure was accurately applied, but also that the collected data presents experimental repeatability. Since these values are so small, we opted for not including the error bars in the referred graphs aiming for better clarity.
- PCA analysis: results in table 1 are shown in normal and scientific notation, making every comparison difficult. The Authors should describe how they evaluated the ppm concentration (starting from the solvent volume) and the sensor resolution, all quantities that compare in Table 2. PCA score plot does not show “well defined” regions; in fact, the Authors analyzed each PC role to understand that PC1 distinguishes the different analytes and PC2 can somehow evaluate a sort of sensitivity. The Authors must claim the difficulties in evaluating the VOC concentrations.
Answer: Answer: Thank you for the suggestions. The mentioned table was improved. Regarding the calculation of the ppm concentrations, additional information was added to the manuscript. In summary, the evaporated volumes can be directly related with the concentration levels through the ratio between the mass of analyte and the mass of air in the chamber. The mass of each analyte was calculated for each volume through their density. The same approach was employed to calculate the mass of air existent in the interior of the 58 L volume chamber. Once calculated both the analyte mass for each volume and the total mass in the interior of the chamber, the corresponding concentrations were estimated by the mentioned ratio. Finally, the concentration levels were converted to ppm scale for easier comprehension. An additional table summarizing the evaporated volumes and respective concentrations was included in the manuscript. Relatively to PCA results, one can define different regions in the PCA score plot, figure 3a) and this plot can be used to identify the type of VOC. But the most interesting of our results is the analysis of the PC1 and PC2 components. PC1 can indicate the type of VOC and the PC2 indicate the concentration. The manuscript was improved in accordance with the suggestions.
- Resolution: the reported values are state-of-the-art, compared with other results, but the Authors do not emphasize this result and must explain how they found these values (see also point 4 and 5).
Answer: Thank you for alerting us about the ambiguity of this paragraph. In fact, this result was poorly addressed in the original version of the manuscript. Further details regarding the calculation of the resolution were included in the manuscript. We hope that the added information meets your expectations.
- Figure 1: several legends are not written in English and use different styles (spaces, uppercase, normal/bold text), same for axis labels. uL should be µL. Figure IVa and b show different isopropanol concentrations.
Answer: Thank you for alerting us about these issues. Overall corrections were implemented in the figures.
- Figure 2: axis labels have different styles. For b, c, and d left scales, it is impossible to identify the minimum step.
Answer: Thank you for alerting us about these issues. Overall corrections were implemented in the figures.
- Figure 3: legends should use the same nomenclature as in previous figures, i.e., Ethanol and not EtOh.
Answer: Thank you for alerting us about these issues. Overall corrections were implemented in the figures.
- English quality must be improved, and several typos removed.
Answer: Thank you for your suggestion. Overall corrections were implemented throughout the manuscript. We would like to thank you once more for all the helpful suggestions. They represent, without any doubt, valuable improvements to our work. We hope that this final revised version of the manuscript meets your quality standards.

Reviewer 3 Report
Overall a very nicely written paper. Easy to read and follow the results.
A couple of very minor points that should be corrected.
Line 110 graphene (best) graphenes (plural ok) no apostrophe
Figure 1 ab,2b,2c acetic acid, ethanol methanol all need correcting
Figure 3 b - space before (
Figure 3 1 and c PC2. is inconsistent and should be PC2-
Otherwise great job
Author Response
We thank to the Editor and to the Reviewers for the comments and suggestions that we think are properly addressed in the present form of the manuscript. We think that our address to these comments, prompted by this review, improved its overall quality.
Reviewer #3
Overall a very nicely written paper. Easy to read and follow the results.
A couple of very minor points that should be corrected.
Answer: Dear reviewer, we would like to thank you for your kind and fair evaluation of our most recent work. We are very happy that our goals, procedures, and results are clearly explained in the document. Thank you for all your suggestions. We addressed and implemented all your recommendations; they represent an evident improvement for our work.
Line 110 graphene (best) graphenes (plural ok) no apostrophe
Answer: Thank you for alerting us about this issue. Corrections were implemented.
Figure 1 ab,2b,2c acetic acid, ethanol methanol all need correcting
Answer: Thank you for alerting us about the figures. Overall corrections were implemented to fix the language, font and captions.
Figure 3 b - space before (
Answer: Thank you. Corrections were implemented.
Figure 3 1 and c PC2. is inconsistent and should be PC2-
Answer: Thank you for alerting us. Corrections were implemented.
Otherwise great job
Answer: We hope that this final revised version of the manuscript meets your quality standards.

Round 2
Reviewer 1 Report
The authors of the manuscript entitled " Graphene Oxide Thin Films for Detection and Quantification of Industrially Relevant Alcohols and Acetic Acid" made the revisions as requested.
In this situation, the manuscript may be published in its present form.

Author Response
We thank to the Editor and to the Reviewers for the comments and suggestions that we think are properly addressed in the present form of the manuscript. We think that our address to these comments, prompted by this review, improved its overall quality.
Reviewer 1
"The authors of the manuscript entitled " Graphene Oxide Thin Films for Detection and Quantification of Industrially Relevant Alcohols and Acetic Acid" made the revisions as requested.
In this situation, the manuscript may be published in its present form."
Answer: Dear reviewer, thank you for your very useful suggestions and thank you for approving the publication of our most recent work.

Reviewer 2 Report
The Authors revised the paper according partially to my previous comments. They should address some crucial issues:
1. Resolution: The Authors refer to Ref.52 for the resolution evaluation. I suppose they evaluated the slope and y-intercept from “the plot of the logarithm of resistance versus the logarithm of concentration”, as described in Ref.52. However, they never show these logarithmic plots. The Authors present resolutions at least one order of magnitude lower than other previous values; thus, they must convince the reader about the consistency and reliability of these resolutions by adding other details.
2. Fig.2a/c have a different axis style from Fig.2b/d.
3. Fig.2b: impedance angle at the lowest ethanol concentration is not at zero volume
4. Fig 2c: bottom axis is in the wrong vertical position.
5. Fig.2b,c,d: impedance unit should be degrees (°), not rad.
6. Fig.3c: left axis label is partially missing.
Author Response
Response to the Comments
We thank to the Editor and to the Reviewers for the comments and suggestions that we think are properly addressed in the present form of the manuscript. We think that our address to these comments, prompted by this review, improved its overall quality.
Reviewer 2
"The Authors revised the paper according partially to my previous comments. They should address some crucial issues:"
Answer: Dear reviewer, thank you for your useful suggestions and for alerting us about some minor issues that were not fixed during the previous round of revisions. Every question/suggestion were now addressed to improve the final version of the manuscript.
- Resolution: The Authors refer to Ref.52 for the resolution evaluation. I suppose they evaluated the slope and y-intercept from “the plot of the logarithm of resistance versus the logarithm of concentration”, as described in Ref.52. However, they never show these logarithmic plots. The Authors present resolutions at least one order of magnitude lower than other previous values; thus, they must convince the reader about the consistency and reliability of these resolutions by adding other details.
Answer: Thank you very much for this important question. In fact, as the present sensor is linearly dependent of concentration and not of the logarithm of the concentration, therefore, the fitting was done using the principal component 2 (PC2) values versus the concentration in ppm (C). The sensitivity (S) was calculated by the slope of the straight line, . Therefore being the uncertainty of sensitivity given also by the fitting. The resolution corresponds to the value calculated . This information was included in the manuscript.
- Fig.2a/c have a different axis style from Fig.2b/d.
Answer: Thank you very much. All the corrections have been done.
- Fig.2b: impedance angle at the lowest ethanol concentration is not at zero volume
Answer: Thank you very much. The correction has been done.
- Fig 2c: bottom axis is in the wrong vertical position.
Answer: Thank you very much. The correction has been done.
- Fig.2b,c,d: impedance unit should be degrees (°), not rad.
Answer: We were experiencing some issues with the word document. Some figures lose their format sometimes. The correction has been done, thank you.
- Fig.3c: left axis label is partially missing.
Answer: We were experiencing some issues with the word document. Some figures lose their format sometimes. The correction has been done, thank you.
